# It is beyond remuneration: Bottom-up health workers' retention strategies at the primary health care system in Tanzania

Nathanael Sirili[1]*, Daudi Simba[2]

**1** Department of Development Studies, Muhimbili University of Health and Allied Sciences, Dar es Salaam, Tanzania, **2** Department of Community Health, Muhimbili University of Health and Allied Sciences, Dar es Salaam, Tanzania

* nsirili@yahoo.co.uk

## Abstract

Although Tanzania is operating a decentralized health system, most of the health workers' retention strategies are designed at the central level and implemented at the local level. This study sought to explore the bottom-up health workers' retention strategies by analyzing experiences from two rural districts, Rombo and Kilwa in Tanzania by conducting a cross-sectional exploratory qualitative study in the said districts. Nineteen key informants were purposefully selected based on their involvement in the health workers' retention scheme at the district and then interviewed. These key informants included district health managers, local government leaders, and in-charges of health facilities. Also, three focused group discussions were conducted with 19 members from three Health Facility Governing Committees (HFGCs). Qualitative content analysis was deployed to analyze the data. We uncovered health-facility and district level retention strategies which included, the promotion of good community reception, promotion of good working relationships with local government leaders, limiting migration within district facilities and to districts within the region, and active head-hunting at training institutions. Retention of health workers at the primary health care level is beyond remuneration. Although some of these strategies have financial implications, most of them are less costly compared to the top-bottom strategies. While large scale studies are needed to test the generalizability of the strategies unveiled in our study, more studies are required to uncover additional bottom-up retention strategies.

## Introduction

The retention of the skilled health workforce in primary health care facilities especially in rural areas has continued to be a frustration to many health systems for decades [1, 2]. While the global population is halved between urban and rural, the health workforce is asymmetrically distributed. Only 38% of nurses and 24% of the physicians serve the rural population [3]. Several studies have pointed out poor working conditions, poor remuneration, poor social services, poor supervision from immediate supervisors, salary delays, lack of accommodation,

**Data Availability Statement:** All relevant data are within the paper and its Supporting Information files.

**Funding:** This study was funded by the Muhimbili University of Health Sciences through the Swedish

International Development Cooperation Agency – Sida under bilateral agreement 2009-2014 as small grant support for faculty development.

**Competing interests:** The authors declare that there is no conflict of interest regarding the publication of this paper

limited opportunities for career growth and limited alternative sources of income as the major perpetuating factors for poor retention of the health workforce in rural areas [4–6].

Tanzania, like many other countries, is struggling with the retention of the skilled health workforce in its primary health care facilities in rural areas [7]. Estimates show that by 2015, Tanzania had less than half the required health workforce with a rural-urban divide favoring urban areas [8]. The shortage of health workers in rural areas jeopardizes accessibility of health care services to rural communities thus contributing to poor health indicators in the country [9]. One of the major efforts in addressing health workforce retention challenges is the 1990s health sector reforms. The health sector reforms aimed at, among other things, increasing health workforce availability through training, absorption and retention [10, 11]. While there have been significant achievements for training, absorption and retention have remained low due to budget limitations [12–14].

Following the adoption of the 1990s health sector reforms, Tanzania operates a decentralized health system. Under the decentralized system, since the mid-1990s districts were made the focal points for health care services planning, provision, and program implementation [15]. As such, they were given the mandate to recruit, deploy and retain health workers which was initially a role of the central government [16]. The administrative reforms also involved the engagement of the community in decision making on matters relating to their health and health care. Therefore, to ensure adequate and systematic community engagement, HFGCs were formed at each health facility. The HFGC is comprised of five members from the community and three appointed members (the health facility in-charge, a member from the village government committee, and a member of the ward development committee). Some of the key roles of the HFGCs are (i) to develop the plans and budget for the facility and (ii) to liaise with Health Facilities Management Teams (HMTs) and other actors to ensure the delivery of quality health services [17]. HFGCs are therefore vital in the retention of health workers at the facilities and districts at large.

In 2006, it was realized that districts were less efficient in the recruitment, deployment, and retention despite the mandate given to them. Therefore, the government revised the decentralization system and formed a partial centralized system, which combined some elements of decentralized and centralized approaches [16]. In the partial centralized system, districts were left with the responsibility of identifying vacancies for employing health workers and retaining the workforce after the vacancies are filled by the central government [16].

In addition to the administrative reforms, the government of Tanzania has tried several other strategies by combining both financial and non-financial incentives to retain the skilled health workforce in rural areas with little or no observable improvements [12]. Financial incentives have often failed to be implemented due to budget limitations while most of the non-financial incentives have ended in budgetary implications and thus also failed to bear the intended results [18]. Even though the central government is responsible for the payment of salaries of health workers employed at the district, other financial and non-financial incentives are the responsibility of the districts. The latter has resulted in a variation in incentives provided to health workers across districts due to the financial capacity of the respective districts and the way priorities are set [19]. Urban districts are economically better positioned and often have better incentives compared to rural districts which have limited opportunities for earning extra income within the professional practice [19].

The popular non-financial incentives across the districts have included among other things; opportunities for continuing education and career development, proactive staff recruitment, compulsory community service, bonding schemes, contracting arrangements and provision of accommodation [15, 20, 21]. However, non-financial incentives like the opportunity for further education is a challenge to the district because for some cadres, providing staff with

opportunities for further education is offering them an exit opportunity because the new quali-fications that will be acquired are not accommodated in the district staffing level [22]. Further-more, most of these strategies have been proposed through the central government and mostly without being piloted, and have been imposed on the districts for implementation. Lack of dis-trict engagement has often resulted in a mismatch between the devised strategy and the resources needed for its implementation, poor understanding of the strategies and lack of ownership of the strategies [15].

In supporting the implementation of these strategies, some development partners have sup-ported some rural districts in improving the retention of health workers through different measures. In southern Tanzania, Deutsche Gesellschaft für Internationale Zusammenarbeit (GIZ) has been implementing the pro-active staff recruitment programs in six districts. In this system, students are enrolled in training institutions after signing a bond with the council. Another organization, the Benjamin Mkapa HIV/AIDS Foundation, BMAF (a local Non-Gov-ernmental Organization), is supporting the construction of houses for health workers in differ-ent districts, contractual employment of health workers in rural districts, and contractual employment of tutors in allied health training institutions. The health workers are contracted to work for two years in remote areas, with a relatively better incentive package compared to that provided by the local government. These health workers are supposed to be employed by the local government authorities at the end of their contracts. While construction of houses has not faced many challenges, the contractual employment has often resulted in challenges once the contract expires as the remuneration provided by the government has always been far lower compared to what was offered by BMAF.

Therefore, despite the reforms, the revision of the administrative system and the different strategies adopted for the retention of health workforce, most of the rural districts have failed to retain an adequate number of skilled health workforce [7, 12, 16]. Furthermore, like in many other parts of the world, the majority of studies on the health workforce issues con-ducted in Tanzania have focused on documenting shortage in terms of numbers, geographical imbalance and the associated factors in general [6, 7, 12, 23]. Previous studies conducted in rural districts of Tanzania have shown that the retention of health workers is influenced by other factors including family ties, socio-cultural factors and availability of social services [10, 22, 24]. They have also shown that little attention has been paid to retention strategies coming from the rural districts to inform the national level. Thus, this study sought to explore bottom-up health workers' retention strategies by analyzing experiences from two rural districts of Rombo and Kilwa.

## Materials and methods

### Study design

A cross-sectional exploratory qualitative study was conducted in two districts of Kilwa and Rombo in 2015–16. This design was considered appropriate because the retention of health workers in rural areas is influenced by social processes and is less studied. Moreover, exploring the bottom-up retention strategies required in-depth exploration while taking into account the context of the districts.

### Study setting and selection of study site

The two regions of Lindi and Kilimanjaro respectively were selected for the following reasons; Lindi is one of the two regions forming the southern zone where proactive staff recruitment, a form of bottom-up approach retention strategy is being implemented. This is among the zones that have a critical shortage of health workers [14]. Kilimanjaro is one of the three regions

**Table 1. Geopolitical zones of Tanzania.**

| Zone | Regions |
|---|---|
| Central zone | Dodoma and Singida |
| Eastern zone | Coast, Dar es Salaam and Morogoro |
| Lake zone | Kagera, Mara, Mwanza, Shinyanga, Simiyu and Geita |
| Northern zone | Arusha, Kilimanjaro, Manyara and Tanga |
| Southern zone | Lindi and Mtwara |
| Southern highlands zone | Iringa, Mbeya, Ruvuma, and Njombe |
| Western zone | Katavi, Kigoma, Rukwa and Tabora |

forming the northern zone, and is among the zones that have a relatively better health worker density compared to other zones, despite that the zone is still below the recommended World Health Organization (WHO) standard of doctors to population ratio of 1: 10,000> [14]. The two zones (Table 1) were purposefully selected to explore the influence of the different cultural practices, economic activities, and the availability of social services. In terms of districts, Kilwa District, one of the six districts forming Lindi region was randomly selected from among five districts of the southern zone where proactive staff recruitment is implemented. Other districts where proactive staff recruitment is implemented are Nachingwea, Tandahimba, Masasi and Nanyumbu in Mtwara region. Administratively, Kilwa is subdivided into 20 wards. Kivinje district hospital plus three randomly selected health facilities from three villages in two randomly selected wards with health facilities were involved in this study.

Rombo district on the other hand, was randomly selected from among five rural districts of Kilimanjaro region and Huruma hospital (District Designated Hospital) plus three other health facilities from three villages in two randomly selected wards with health facilities were involved in this study.

## Selection and recruitment of study participants

We used purposive sampling to identify key informants and focus group discussion participants by their positions and responsibilities with regard to health workers' retention. Based on our experience and knowledge of the local government administrative structure in Tanzania, we started by listing all people who are involved in human resources management activities at the selected districts. In this regard, we identified and included the District Executive Director (DED) as the main employer, and Human Resources Officers (HROs) as technical officers to the DED on matters related to human resources management. We identified and included other local government executives at the ward and village level (Ward Executive Officers–WEOs and Village Executive Officers—VEOs) who are the technical advisors of the DED at the village and ward, and also form part of the development committees at those levels. At the health department, we included health managers (District Medical Officer–DMO and District Health Secretaries—DHS) who are the immediate supervisors of health workers at the district. We also identified and included the day-to-day supervisors at the health facilities (District Hospital Medical Officers in-charge, Health Centers medical in-charge and dispensaries Clinical Officer-in-charge).

Furthermore, we identified and invited members of the HFGCs. The latter are involved in budgeting, planning, and decision making around the management of the health facility and thus play a vital role in the retention of health workers. We identified these through the in-charge of the respective health facilities in collaboration with village and ward leaders and they were invited to participate in the focus group discussions.

## Data collection

Using a semi-structured interview guide and focus group discussion guide written in Kiswahili language, we conducted 19 key informant interviews and three focus group discussions with 19 members of HFGCs (Table 2). A trained research assistant with a background in social science and more than ten years' experience in conducting qualitative research was taking notes during the interviews and discussions as they were being conducted by the researcher. All KIIs and FGDs were audio-recorded using a digital voice recorder. The audio recorder was kept in a secure place by the researcher. The KIIs lasted between 30 and 60 minutes while the FGDs lasted between 60 and 90 minutes. The interviews were carried out in the informants' offices or designated rooms at the health facilities. Each FGD comprised 6 to 7 members of the HFGCs and were conducted at one of the designated rooms at the health facility or classroom in a nearby school as identified and arranged by the village leader.

## Data analysis

All KIIs and FGDs were first transcribed verbatim. Qualitative content analysis as explained by Graneheim and Lundman [25] was used in the analysis of the data. This involved reading and re-reading the transcripts to get familiar with the data. We adopted a hybrid approach in which both inductive and deductive analysis were conducted. A deductive approach was used to develop a preliminary codebook by the first author based on the objectives of the study and familiarity with the data. On the other hand, inductive approach was used to add new codes that emerged from the data during analysis. The codebook was discussed with the second author and after agreement, the codebook was imported to NVIVO version 11 software before the coding. All transcripts were treated as meaningful units. Condensed meaning units were then identified and coding was done with the aid of the NVIVO software. The coding was done by the first author and all codes were discussed and finally agreed upon by both authors. The coding process was iterative and new codes were identified that either supplemented the prior codes or were added as new codes. Then, similar codes were grouped and then abstracted into sub-categories. Based on similarities and differences between the sub-categories they were further grouped into categories to reflect both latent and manifest content. In this analysis, the DED, WEO, and VEO are considered as Local Government Executive Officers whereas the DMO, DHS, and DHRO are considered as Health Managers.

## Ethical considerations

Ethical clearance for this study was obtained from the Senate Research and Ethics Committee of the Muhimbili University of Health and Allied Sciences in Tanzania (Ref. No. 2015-01-15/

**Table 2. Participants for KIIs and FGDs.**

| Category | Male | Female | Total |
|---|---|---|---|
| *Key Informants* | | | |
| District Executive Director | 1 | 0 | 1 |
| District Medical Officers | 2 | 0 | 2 |
| Ward Executive Officers | 1 | 1 | 2 |
| Village Executive Officers | 1 | 1 | 2 |
| District Health Secretaries | 0 | 2 | 2 |
| Health Facility In-charges | 4 | 5 | 9 |
| Human Resources Officer | 0 | 1 | 1 |
| *Focus Group Discussion participants* | 11 | 8 | 19 |

AEC/Vol. IX/43). Permission for data collection was obtained from the Ministry responsible for health, Ministry responsible for local government, regional and district authorities, and Heads of health facilities. Written informed consent was obtained individually from each study participant before the commencing of the interview or the discussion. The objective and importance of the study were explained to study participants and they were encouraged to respond transparently as information from them will only be used for research'purpose. They were further informed that their participation was purely voluntary and they can withdraw from the interview or discussion any time they wish to do so with no repercussion. Confidentiality of the information was ensured by avoiding mentioning names or titles of the key informants or FGD participants during recording. FGDs participants were assigned numbers that were kept separately from the audio record descriptions. To ensure privacy typed notes and audio records were kept in a pined folder in a computer of the first author that was only accessible to him. Only the transcripts were shared with the second author.

## Results

From 19 KIIs and 3 FGDs with 19 participants, two categories of retention strategies emerged, the health-facility level and the district-level health workers retention strategies (Table 3).

### Health facility level health worker retention strategies

**Promoting good community reception.**   Good community reception to incoming health workers was reported by some health facility in-charges as an important contributory factor to the retention of health workers. Health facility in-charges reported that they organize small functions in collaboration with local government leaders to introduce new staff to the community. Sometimes the function includes a tour around households within the community in the company of the respective community leader/s. This makes the newly employed health workers feel valued by the community they serve and thus more likely to be retained.

> "...once a new health worker has reported to our health facility, the in-charge informs us and together we organize a day when we officially welcome and introduce that health worker to our community during a small party...When that is not possible, one of us (a member of health facility governing committee), plans a day to show the health workers around and introduce them to the community..." [FGD member-6- District A].

**Enhancing a good working relationship with local government leaders.**   In Rombo district, local government leaders reported working closely with health workers and participating

**Table 3. Summary of retention strategies by level.**

| Category | Retention strategies |
|---|---|
| Health-facility level health worker retention strategies | • Promoting good community reception<br>• Enhancing good a working relationship with local government leaders<br>• Promoting good cooperation and trust among health workers<br>• Prize giving to the best performer of the year |
| District-level health worker retention strategies | • Provision of career opportunities and support<br>• Limiting migration within district facilities and to districts within the region<br>• Provision of financial incentives<br>• Support from the Non-Governmental Organizations<br>• Efforts by Health Facility In-charges in resolving housing problems<br>• Active headhunting in training institutions |

in various social activities together. This encourages health workers to feel valued and develop a sense of belonging. They added that the latter made them feel satisfied with their work as well as their working environment. Informants further reported that issues related to the day-to-day running of health facilities were discussed regularly in meetings organized by local government leaders in which the in-charges of health facilities are members.

*". . .the in-charges of health facilities are members of our Ward Development Committees and Village Development Committees. . .we meet at least every month to discuss various issues including the challenges facing our health facilities. . ."* (Local Government Executive Officer-2-Rombo)

**Promoting good cooperation and trust among health workers.**   Good cooperation and trust among health workers in the same health facility was reported to promote the retention of the health workers. In Kilwa for instance, mutual trust was reported to be high to an extent that the health workers exchanged bank cards and passwords and only one person would go to the bank to withdraw money for their colleagues. This mutual trust was said to contribute to the retention of health workers.

*". . .health workers here trust one another. . .sometimes those who are in remote areas appoint one person (to go to town) . . . to draw their salaries for them. . . they give the person their ATM cards with pin code . . . they contribute some cash for transport and lodging costs . . . . . ."* (Health manager-2-Kilwa)

**Rewarding the best performer of the year.**   In Rombo, we were informed that some facilities had made internal arrangements in which the best performing staff were given prizes at the end of the year. Health workers contribute some cash to facilitate annual parties. In these gatherings, new-comers are invited and introduced to other members of staff and local government leaders while those retiring or transferred to other places are bid farewell. Participants added that these gatherings and collective rewarding made them feel a sense of belonging to the community and thus influenced their retention.

*". . .we contribute among ourselves some cash not exceeding twenty thousand (equivalent to eight US Dollars), and organize a party every year. . .in this party, we welcome and introduce newly employed health workers in that year, we introduce them officially to the community, we bid farewell to retirees and transferred workers . . . we give prizes to the best performers of the year. . .all these make our workers feel valued and united. . ."* (Health facility in charge-6-Rombo)

### District level health worker retention strategies

**Provision of education career opportunities and financial support.**   Participants from the two districts revealed that the district authorities had devised and were providing two different forms of education career opportunities and support. In Kilwa, the district authorities have identified priority areas for educational career support and only those who aspired to progress in those areas were supported. They further added that priority was given to cadres with severe shortage of staff. One informant affirmed that this strategy has contributed to the retention of some health workers.

*". . .I have just announced to nurses that those who want to undertake training on anaesthesia at KCMC or Muhimbili or Ifakara should inform the management in writing and we will*

*support them. . .for instance, we offered financial support to one Clinical Officer who was an Assistant Clinical Officer . . . he has just returned . . ."* (Health manager-1-Kilwa).

In Rombo, participants mentioned the existence of an organized rotational plan which indicates who should go for training and when. The plan also indicates who will provide back-up for the one going for training. One informant reported that after coming back from training he was not posted back to where he used to work but rather to another facility to allow the one who was serving there to go for studies. Participants further added that the rotational plan has helped to curb staff shortages and provide equal opportunity for educational advancement and career growth to all staff.

*". . .When I was released for studies another person was sent to my station when I returned, I was brought to this facility and the one who was in charge here went for studies. . .As we speak now, he is in his second year of training. . ."* (Health facility in charge-2-Rombo).

**Limiting migration within district facilities and to districts within the region.** Participants from Kilwa reported that in recognition of the serious shortage of health workers in the district, the Regional Commissioner directed district authorities to limit migration of health workers within the region, except where the reason for migration outside the district are justifiable. Regarding this directive, authorities in Kilwa reported scrutinizing requests for migration. In most cases, permission to migrate is given within the district and on very rare occasions to other districts, within the region. Managers felt that limiting migration within districts helped to minimize overall migration of workers.

*". . .Ooh yes, this strategy of limiting migration within the district and rarely within the region, has helped us to reduce migration of the few staff we have to other regions"* (Health manager-2-Kilwa).

**Provision of financial incentives.** Although district authorities clearly stated that the districts were facing serious constraints due to a shortage of funds, they both admitted the importance of providing financial incentives to health workers as a retention strategy. In the two districts, on-call allowances and the provision of other financial incentives were reported to be affected by the scarcity of funds. We were further informed that district health managers were putting up measures to solicit funds for the provision of financial incentives to health workers.

In Rombo, health managers were struggling to ensure that at least all newly employed health workers get their subsistence allowances immediately after reporting before they receive their first salary. In Kilwa, health managers reported negotiating with district authorities to use internally generated funds to provide financial incentives to health workers. They added that the provision of financial incentives helps in the retention of their health workers.

*". . .We have spoken to the district authorities and we have requested that they allocate 10% of internally collected revenue for providing incentives to the health workers. . .we are waiting for the budget of the coming financial year to see if this will be implemented. . ."* (Health manager-2-Kilwa)

**Support from non-governmental organizations.** Another strategy to facilitate health workers retention that was mentioned by authorities in both districts was to work in collaboration with the Non-Governmental Organizations (NGOs) in the construction of staff housing and outreach services.

In Kilwa District, the housing construction strategy received support from the Benjamin Mkapa Foundation (a local NGO), where they worked together in the construction of staff houses. Informants stated that the constructed houses, especially in rural areas, have helped to retain new staff who were posted there and has made a big difference when compared to when there were no houses.

*". . .previously, it was a big challenge, you post a new worker to a village and you find villagers carrying grass for thatching a house for them . . .the next day they will be at your office with a hundred reasons seeking to migrate. . .so we sat down and requested the Benjamin Mkapa foundation to help us to build houses in rural areas, it is on-going and we hope when this is completed the migration rates will go down. . ."* (Health manager-1-Kilwa)

**Efforts by health facility in-charges to resolve housing problems.**   In Rombo, although no NGO was reported to directly support the construction of houses, they were supporting other outreach programs. For housing construction in Rombo, the health facilities in-charges reported taking active roles in solving housing problems. They reported making efforts to convert available buildings that belonged to the health facility into a habitable place for incoming staff. The provision of housing is said to influence the retention of health workers.

*". . .what you see as a new house was an old unused dispensary building, I have struggled to solicit funds to renovate it and partition it to become a residential house, two families are staying there now. . ."* (Health facility in charge-3-Rombo)

**Active headhunting in training institutions.**   Another retention strategy in Kilwa was implemented by district authorities who visited health training institutions to talk to final year students and inspire them to go to Kilwa for attachment while waiting for their final examination results. This strategy aimed to get students and potential recruits to familiarize themselves with the Kilwa environment. Managers felt that when the students are familiar with the working environment it will promote their retention if posted to Kilwa as their work station.

*". . .Because of the pressing shortage of health workers in our district, we request some influential people from our district to go to training institutions and talk to final year students to inspire them to come and work in Kilwa. . . we provide some allowances to support the stay of those who agree to come and immediately the recruitment posts are announced we collect their certificates and request the ministry to post them to Kilwa as they are in Kilwa already. . ."* (Health manager-1-Kilwa)

## Discussion

We aimed to explore bottom-up strategies for the retention of health workers at the primary health care system by conducting our study in two rural districts of Rombo in the northern zone and Kilwa in the southern zone of Tanzania. In this study, we report two major categories of bottom-up health worker retention strategies; (i) Health facility-level health worker retention strategies which are the promotion of good community reception to newly employed health workers, promoting a good working relationship with local government leaders and good cooperation and trust amongst health workers, and the rewarding of best performers of the year; and (ii) District level health worker retention strategies which include limiting migration of health workers within regions, working in collaboration with Non-Governmental Organizations in housing construction, active headhunting at training institutions, and the

provision of education advancement and career opportunities through financial support. From the two categories, we feel that health worker retention strategies go beyond remuneration. Despite some of the strategies having some overlapping elements, in this section, we start by discussing the health facility level strategies and end-up with district level strategies. We discuss them distinctly as the two involve different levels of players.

The affiliation of new-comers to the community and work environment as revealed by our study brings a sense of belonging and acceptance of health workers by the community. Sirili et al. 2018 reported that poor working relationships amongst health workers and between health workers and the community through their local government leaders contributes to poor retention of health workers in rural areas [19, 22]. According to Maslow's hierarchy of needs, a sense of belonging and acceptance create a sense of satisfaction [26]. Several studies from different contexts have shown that job satisfaction is linked with the retention of workers [27–29]. Therefore, having strategies that impact on the satisfaction of health workers is important, however, care must be exercised in the so-called '*good community reception*' as different ethnic groups may have a different interpretation of what is good or bad especially when the receiving community and the workers are from different ethnic groups.

The provision of prizes to best performers of the year as devised by some of the facilities is perhaps a less costly strategy and can be improved by institutionalizing it such that the incentives come from health facility resources rather than from individual contributions. It is strongly advised that district authorities adopt this strategy and scale it up. By scaling up the strategy to the district level, district authorities can set harmonized packages for different levels of facilities for both facility and individual-level performance. We feel that the latter will ensure the sustainability of this strategy. The WHO in its global policy recommendations, states that appropriate financial incentives are an important retention strategy [3], however, to implement financial incentive strategies sustainably, health facilities must devise innovative approaches for income generation. One way practiced by some urban health facilities is the opening of a window for intramural private practice services after the normal working hours for selected services [19]. This may help health facilities to generate some income for providing certain financial incentives like extra-duty allowances, responsibility allowances and even rewarding performance at the end of the year. However, its practicability may be challenged in some facilities especially those in rural areas.

At the district level, the study unveiled limiting migration of health workers within regions, working in collaboration with Non-Governmental Organizations in housing construction, active headhunting at training institutions, and the provision of career opportunities and career support as the most prominent strategies. Limiting the migration of health workers to facilities within the district or to other districts within the region as found in this study is a managerial decision and perhaps a less costly strategy in comparison to most of the other identified strategies. This strategy is crucial as it taps into political will and enhances the understanding that health worker retention goes beyond the health sector. The latter is pointed out by several other studies in different places [30–33]. However, the working environment within the district varies from one facility to another and thus without careful measures to regulate movement within the district, some facilities will continue to be more affected than others. Furthermore, migration of health workers from one place to another is a product of many drivers including family ties [19, 22], which may limit a person from moving out of the district or region as it may have both social and economic consequences and thus result in dissatisfaction. Studies have shown that job satisfaction is directly linked to workers' retention, as such, coercively limiting migration would only have short term results [34]. Therefore, there is a need to develop long term strategies to apply multi-sectoral stakeholder engagement to address

issues around family union, social services, and the working environment at remote areas of the districts in question [35].

Working in collaboration with Non-Governmental Organizations in housing construction as revealed in this study is not different from what other studies have documented [3, 36, 37]. This strategy is in line with both Tanzania's public-private partnership (PPP) policy and the WHO advocacy on the role of collaborative work between the public and private sectors for improved results [38, 39]. The revealed efforts by heads of facilities and community members in building houses for health workers call for joint efforts by all stakeholders to lessen the burden that the government must carry against other competing priorities and limited resources. This strategy not only reduces the resource burden but also creates a sense of ownership by the community of their own health. In places where NGOs can work with facilities and communities to construct houses, such efforts and engagement should be promoted and sustained.

Active head hunting for graduating students in health training institutions as revealed in this study was a strategy used previously in Tanzania when districts had the mandate of employing health workers [40]. Among the weaknesses of this strategy was the failure of some poor districts to attract health workers, however, the strategy has the advantage that those who agreed to go to said districts were more likely to stay [12, 41]. This strategy might work better in this era where there are many graduates competing for limited employment opportunities in the health labour market [14]. To make this strategy more attractive and effective, once these graduates have joined the respective districts, deliberate efforts should be made to employ them immediately recruitment slots are made available by the government.

The enhancement and support of career opportunities by granting permission and the provision of financial support for further education as used in the two districts is widely recognized by many authors as a good strategy for the retention of the health workforce in underserved areas [33, 36, 37, 42]. Even though the providing of educational advancement and career opportunities and financial support, is termed as a non-financial incentive, its implementation needs financing. The provision of education and career opportunities is widely reported as one of the retention strategies for health workers used in rural areas in many places [30, 32, 33]. However, a study by Sirili et al. [22] in 2018, reported that the provision of career opportunities in rural Tanzania was challenged because the organization of health services at district hospitals and the human resources for health manning levels do not provide room for a specialist to be employed at the district level [10]. In this regard, offering the opportunity for specialization as a retention strategy may promote migration from the district and defeat the purpose. Based on these findings, the authors feel that policymakers need to re-examine the use of developing career opportunities as a non-financial incentive for the retention of health workers in rural Tanzania. The re-examination should consider both the policy landscape such as schemes of service available and the resources to implement this strategy sustainably.

In general, the findings of this study like in other previous studies in Tanzania and other places [6, 30, 43–46], underscore the fact that retention of health workers is a complex phenomenon that requires multiple strategies at different levels. We feel that with the growing population and increased burden of non-communicable diseases amidst communicable diseases, the failure to retain health workers in rural areas may lead to worse health indicators than the current status.

## Methodological consideration

We start by discussing how we enhanced the trustworthiness of the study findings. In qualitative studies, the findings are trustworthy if they are worth believing [25]. We adopted Guba's four criteria of credibility, dependability, transferability, and conformability to enhance the

trustworthiness of our study findings [47]. The credibility of the findings of this study was enhanced by the triangulation of informants' perspectives with experiences and rich information on the study questions. To enhance the credibility and dependability of the study, triangulation of study settings and researchers was used. To enhance the conformability of the findings by ensuring that the findings reflected informants' perspectives rather than the researchers' understanding of the question under study, categories were inductively generated and presented with the support of subcategories and quotes. The transferability of the findings of this study is enhanced through the description of the study setting, context, data collection process, and analysis.

The findings of this study are subject to three major limitations. One, the fact that interviews were conducted by a medical doctor (the first author) might have influenced informants to respond in a socially desirable manner. However, this was offset by having research assistants with social sciences background.

Two, only two districts were involved and in a few selected facilities, this may have left out other bottom-up strategies. However, the involvement of different categories of informants added richness to the study and enhanced the applicability and truth value of the findings. Third and last, this was a qualitative study and thus only a few selected informants were interviewed. The latter may have left many other strategies out. However, the involvement of a whole range of informants from the district hospital level to dispensary level and community representatives offset this limitation. Finally, the findings of this study reflect the situation during the period in which data collection for this study was being carried out.

## Conclusion

The findings of this study underscore that retention of health workers at the primary health care level requires not only central (national) level strategies but rather a combination of strategies from the national to the facility level, financial and non-financial, and strategies that target the individual health worker as well as the general community where they work. Bottom-up retention strategies have an element of community, health worker and community participation in governing health facilities and thus are easily understood by stakeholders at the primary health care level. The latter is crucial for ownership thus making the strategies more likely to be sustainable. Finally, the retention of health workers in rural districts calls for multiple strategies beyond financial incentives. Further studies are needed on a large scale to uncover more bottom-up retention strategies and test the few that were identified by our study.

## Supporting information

**S1 Transcripts.**
(DOC)

## Acknowledgments

The authors sincerely acknowledge the support from the Ministry of Health, Community Development, Gender, Elderly and Children and the local government authorities and all research participants.

## Author Contributions

**Conceptualization:** Nathanael Sirili, Daudi Simba.

**Data curation:** Nathanael Sirili.

**Formal analysis:** Nathanael Sirili, Daudi Simba.

**Funding acquisition:** Nathanael Sirili.

**Methodology:** Nathanael Sirili, Daudi Simba.

**Writing – original draft:** Nathanael Sirili.

**Writing – review & editing:** Nathanael Sirili, Daudi Simba.

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
