## [Decision Letter · Decision Letter 0]

23 Jul 2020

PONE-D-20-09192

It is beyond remuneration: Bottom-up health workers’ retention strategies at the primary health care system in Tanzania

PLOS ONE

Dear Dr. Sirili,

Thank you for submitting your manuscript to PLOS ONE. After careful consideration, we feel that it has merit but does not fully meet PLOS ONE’s publication criteria as it currently stands. Therefore, we invite you to submit a revised version of the manuscript that addresses the points raised during the review process.<please by="" manuscript="" revised="" submit="" your="">

Please include the following items when submitting your revised manuscript:</please>

We look forward to receiving your revised manuscript.

Kind regards,

Khin Thet Wai, MBBS, MPH, MA (Population & Family Planning Resear

Academic Editor

PLOS ONE

Journal Requirements:

2. Please include additional information regarding the interview guide used in the study and ensure that you have provided sufficient details that others could replicate the analyses. For instance, if you developed a guide as part of this study and it is not under a copyright more restrictive than CC-BY, please include a copy, in both the original language and English, as Supporting Information.

3.We note that you have indicated that data from this study are available upon request. PLOS only allows data to be available upon request if there are legal or ethical restrictions on sharing data publicly. For information on unacceptable data access restrictions, please see http://journals.plos.org/plosone/s/data-availability#loc-unacceptable-data-access-restrictions.

Additional Editor Comments (if provided):

English language editing is necessary to correct few grammatical errors.

Authors need to follow the Consolidated criteria for reporting qualitative research (COREQ): a 32-item checklist for interviews and focus groups and to cite the reference in methods section.

Reviewers' comments:

Reviewer's Responses to Questions

**Comments to the Author**

1. Is the manuscript technically sound, and do the data support the conclusions?

Reviewer #1: Yes

Reviewer #2: Yes

Reviewer #3: Partly

2. Has the statistical analysis been performed appropriately and rigorously? 

Reviewer #1: N/A

Reviewer #2: Yes

Reviewer #3: N/A

3. Have the authors made all data underlying the findings in their manuscript fully available?

Reviewer #1: Yes

Reviewer #2: Yes

Reviewer #3: Yes

4. Is the manuscript presented in an intelligible fashion and written in standard English?

Reviewer #1: Yes

Reviewer #2: Yes

Reviewer #3: Yes

5. Review Comments to the Author

Reviewer #1: This is an interesting article, and important contribution to the literature on the substantial human resources problems encountered in LMICs, and particularly in sub-Saharan Africa. The authors include important information on “bottom up” strategies to improve health care worker (HCW) retention in rural areas, a problem with significant urgency in Tanzania, with such a large rural population and far too few health workers. Because of this emphasis, the authors focus on ways to recruit and retain more HCWs identified by health system officers (DMOs, etc) and facility executive officers, who are “closer to the ground” on these issues and can offer important insights. I think this summary and description of their suggestions is important, and am very supportive of this article being published, with the revisions I suggest below to strengthen this article:

I think the biggest weakness of this article is the lack of much background, explanation, and description of the larger HR problems in the country, which I believe would strengthen this article and make it more widely relevant to the journal’s readership. In a background section perhaps, a bit more information on the HR crisis in TZ, and I think more clearly and expansively explaining why it’s important to survey health workers on retention strategies in a “bottom up” way is important. Because the authors themselves have already contributed a significant amount of excellent scholarship on this topic, I believe this is possibly somewhat easily done. Several of my comments relate to this overarching weakness.

The title is “beyond remuneration” but the authors don’t explain this – was this a quote? Does this reference that the TZ MoHSW doesn’t have funding to provide more adequate remuneration, meaning facilities/districts need to find money elsewhere to provide to HCWs? Do the HCWs themselves consider their current remuneration inadequate? Some more explanation about this, perhaps in a “background” section that more fully outlines the problems of HCW staffing and retention would help to strengthen the article.

Are there any “perks” given by the central gov’t for working in rural areas (outlined briefly on. P. 3) found to be useful or working to retain HCWs? Or is much of the problem because the gov’t doesn't have the funding to support the strategies it tries to implement, as the authors sort of suggest (“often financial incentives have failed to materialize…”? Since the argument is that centralized, top-down strategies are inadequate, a bit more description of what those top-down strategies are would help clarify (related to the description in the intro, on p. 2).

More explanation about the historical process of decentralization and the 2006 creation of a partially centralized system (p. 2- 3) – particularly as all the data presented here sort of toggles back and forth between centralized and decentralized issues (like top-down and bottom-up approaches, which I’m roughly mapping on to that distinction). It would also help to clarify how funding works in a decentralized process – to what degree are regions, districts, and facilities responsible for funding salaries, accommodations, and other incentives the authors describe? In some places the authors write about income generating activities, does this mean they don't receive enough from the gov’t, or need to generate funding from local sources? For some context, see Marten & Sullivan (2020) – Hospital Side-Hustles: Funding Conundrums and Perverse Incentives in the Publicly-Funded Tanzanian Health Sector. Social Science & Medicine 244: 112662.

Regarding the selection of Kilwa and Rombo – why do the two zones have different health worker shortages? Do they have different issues? Did Rombo experience fewer issues of HCW shortage, and why? Does Kilwa have bigger problems, and why? I realize the authors have previously written about this, but a very short description of the background here would help situate the results.

Some lines need a bit more clarity and explanation:

-P. 3, “in the partial centralized system…” unclear the meaning of this sentence

-P. 3, “Most of these strategies…and …this has created challenges like poor understanding…” more explanation of this would help readers better understand the need for more bottom-up strategies like the authors propose

-P. 4, unsure what Kilimanjaro’s “better health workers’ availability” means

-P. 5, “natural settings of the informants” sounds awkward, perhaps just something like “the interviews were carried out in informants’ offices or in designated rooms at their facilities” ?

-P. 8, the subhead “enhancing good working relationship…” – more explanation of why this might work

-P. 8, “fostering good cooperation” – this seems like an outcome, rather than a strategy that could be implemented – did the informants describe how trust was fostered? A bit more explanation here

-P. 9, in “district level” subhead “provision of career opportunities: “inclined to the cadres” is unclear; clarify what support means, as it is frequently mentioned but is a somewhat vague term that could likely mean a lot of things –

-P. 10, somewhat unclear about the process of limiting migration of HCWs – how or why do HCWs migrate? Why would district authorities have the power to migrate HCWs outside a district or region? A bit more detail here –

-

Not sure Table 1 is necessary? Not sure it adds much to the paper.

Some explanation either in the subheadings under Results, or in the discussion, of whether these strategies were deemed effective (or if informants thought they would be effective) at retaining HCWs, and perhaps scalable?

p. 11 – related to some of the points outlined above, regarding more background explanation of the HR for health crisis in Tanzania, more description about what subsistence allowances are, HCW salaries, etc. I know that some changes in salaries, pensions, health insurance, etc. have been implemented recently in TZ to address HR issues, it would be helpful to know more about these changes, (in a background section) and if HCWs themselves believe their salaries to be adequate?

p. 15 – In the discussion, it appears as though some of the burdensome consequences of decentralization (asking local communities to fund services) and neoliberalism/austerity are being supported by the authors (which is related to why they didn’t work effectively)? I disagree somewhat with the assessment that local communities need “to lessen the burden that the gov’t must carry”; as a newly lower middle-income country, the TZ gov’t has more money than the rural poor to pay user fees, etc. Or is the argument that NGOs should support this more? Unclear which stakeholders are asked to shoulder this burden…

p. 15 – “this strategy might work better in this era where there are many graduates…” – can the authors clarify this a bit – some more information about changes in the numbers of students graduating, hiring practices by the central gov’t, salary lines available, etc. to reduce the HR crisis? Why aren’t all grads employed if there is a HR crisis? Some reference to financial constraints here would be helpful, or maybe added to a background section

p. 16 – “the provision of career opportunities” – provide an example to clarify what kind(s) of career opportunities are promoted

p. 16 – unclear the two sentences beginning with “however, a study by Sirili et al. 2018…” – not sure what staffing levels at specialized cadres means…some clarification

p. 16 – study trustworthiness should probably go in the methods section

There are some typos and some awkwardly worded sections, like at the very bottom of p. 15 ("although"), p. 2, "poor working condition" should be "conditions", p. 2 "Tanzania, an East African country as for many other countries" is awkward, p. 3, unsure what the "latter" is referencing, unclear, p. 10 "suffocation" should probably just be "the districts were seriously constrained",

Reviewer #2: The topic for the study is relevant in current context where there is a problem of retention of health workforce specially in developing countries. I have few comments to offer:

1. Methodology

- Are the two jurisdictions selected randomly our of poor and well-off jurisdictions from North and South Zone. You may make a table showing different health workforce indicators of two jurisdictions.

-Why only selected numbers of informant interviewed

Overall, the results are nicely presented

2. Discussion

- The discussion can upfront mention the novel key findings in the beginning.

-There is ample literature on retention of health workforce in developing nations, like India, South Africa etc. The author could have mentioned few of them

- The limitations and strengths of the study could have been mentioned.

Reviewer #3: PONE-D-20-09192

Tittle: It is beyond remuneration: Bottom-up health workers’ retention strategies at the primary health care system in Tanzania

Comments

Overall, the manuscript tackles an important area of public health significance looking at how to retain health workers in rural areas to provide the needed healthcare to people. However, the paper requires some major revisions to be acceptable for publication.

Abstract

1.In the method section, it would be important for the authors to briefly state the type of study and also the sampling strategy used to select participants.

Main text

Introduction

1.It would be important to review literature and look at the effects of the problem. If it is difficult to retain health workers in rural areas, what are the effects in terms of health care delivery?

Methods

Study design

1.In the study design section, the authors need to give more explanation to justify why qualitative method was appropriate for this study.

2.The authors also need to state the category of people included in the study

Study site

1.It would be important that the authors provide information on the number of health facilities in the rural districts where the study was conducted and also the number of health workers in the two districts.

2.Are they all located in rural areas or one is in rural area while the other is located in urban area? It is important to provide this information.

3.Also what is the population in the study area?

Recruitment of participants and data collection

1.The statement that reads: “Therefore, District Medical Officers, District Health Secretaries, District Executive Directors, the District Hospital Medical Officers in charge, Health Centers medical in-charge, Ward Executive Officers, dispensaries Clinical Officer-in-charge and the Village Executive Officers” The sentence does not read well and needs some revision.

2.The authors only stated that purposive sampling was used in the study without describing processes or how this sampling strategy was used to select study participants. The authors need to tell the readers how the sampling strategy was used to select participants.

3.Just stating that 3 FGDs were conducted is not enough. The authors need to describe how the FGDs were organized and also the KIIs. How many people constituted one FGD? Were males and female combined?

4.In the data collection section, the authors need to state the number of KIIs and FGDs that were conducted before the table.

5.The authors mentioned that the FGDs were conducted with 19 members of Health Facility Governing Committees. Who are these people and what role do they play at the health facility level and why was it important to include them in the study?

6.The authors did not describe the experience and qualification of those who conducted the interviews? Were they trained and for how long before the interviews were conducted? Was pre-test done to test the interview guides? All these things need to be stated in the method section as well.

Data analysis

1. What strategies did the authors use to arrive on the themes? This need to be stated clearly in the data analysis section.

2.How many people were involved in the coding?

3.Please include the ethics approval ID in the ethical consideration paragraph?

Results

1.I suggest all quotations should be in italic form.

2.In each of the factors mentioned, it will be important to look at views expressed by study participants in the two study districts. Also, are all retention factors reported in the study mentioned in two districts or some were mentioned in one study district while others, also mentioned in the other district. It will be important to look at differences in views in the two districts concerning the retention factors reported.

3.Limiting migration was reported as district level strategy to retain health workers in the study. Is the problem of retention of health workers limited to them moving from the rural area to the urban area or there are instances where some even leave the service and take up other opportunities elsewhere? If taking opportunities elsewhere is one factor, how will limiting migration solve the problem as reported in the study?

4.Though, so many retention strategies were reported in the study, some of them might have been mentioned by only few participants meaning they are less important. I suggest the authors only pick the more important ones and report on them. Other strategies are only temporally e.g Active headhunting at the training institutions.

5.The authors mentioned that both FGDs and KIIs were conducted in the study. Reading through the quotation, I did not see quotations from the FGDs. If quotes from the FGDs were also used then it will be important to differentiate them by labeling the quotes appropriately using KIIs and FGDs throughout the results section.

6.Was frequent visit by health authorities at the district office to health workers working in rural areas mentioned as one of the strategies to retain health workers in Tanzania? Check an article that was published

Discussion

1.In the discussion section, so much emphasis was placed on reports from previous studies to the neglect of the implication of the study results. The authors need to take the key factors/findings of the study and discuss them thoroughly, point out the reasons for the findings and the implications of the findings of the study. What could happen if the situation is not addressed? All these need to come out in the discussion.

2.There is need for proof reading of the manuscript

6. PLOS authors have the option to publish the peer review history of their article (what does this mean?). If published, this will include your full peer review and any attached files.

Reviewer #1: No

Reviewer #2: **Yes: **SONU GOEL

Reviewer #3: **Yes: **Samuel Tamti Chatio

---

## [Author Response · Author response to Decision Letter 0]

2 Nov 2020

The responses in details have been attached with the cover later

---

## [Decision Letter · Decision Letter 1]

21 Dec 2020

PONE-D-20-09192R1

It is beyond remuneration: Bottom-up health workers’ retention strategies at the primary health care system in Tanzania

PLOS ONE

Dear Dr. Sirili,

Thank you for submitting your manuscript to PLOS ONE. After careful consideration, we feel that it has merit but does not fully meet PLOS ONE’s publication criteria as it currently stands. Therefore, we invite you to submit a revised version of the manuscript that addresses the points raised during the review process.Please submit your revised manuscript by Feb 04 2021 11:59PM. If you will need more time than this to complete your revisions, please reply to this message or contact the journal office at plosone@plos.org. Please include the following items when submitting your revised manuscript:

We look forward to receiving your revised manuscript.

Kind regards,

Khin Thet Wai, MBBS, MPH, MA (Population & Family Planning Res.)

Academic Editor

PLOS ONE

Additional Editor Comments (if provided):

Please check and correct the grammatical errors throughout the manuscript which is deemed necessary.

Reviewers' comments:

Reviewer's Responses to Questions

**Comments to the Author**

1. If the authors have adequately addressed your comments raised in a previous round of review and you feel that this manuscript is now acceptable for publication, you may indicate that here to bypass the “Comments to the Author” section, enter your conflict of interest statement in the “Confidential to Editor” section, and submit your "Accept" recommendation.

Reviewer #3: All comments have been addressed

2. Is the manuscript technically sound, and do the data support the conclusions?

Reviewer #3: Yes

3. Has the statistical analysis been performed appropriately and rigorously? 

Reviewer #3: N/A

4. Have the authors made all data underlying the findings in their manuscript fully available?

Reviewer #3: (No Response)

5. Is the manuscript presented in an intelligible fashion and written in standard English?

Reviewer #3: Yes

6. Review Comments to the Author

Reviewer #3: PONE-D-20-09192R1

It is beyond remuneration: Bottom-up health workers’ retention strategies at the primary health care system in Tanzania

Further comments to the authors

1.The authors need to explain in more details how the purposive sampling method was used to select study participants. Mere mention of purposive sampling is not enough. Please, state how participants were selected for the KIIs and the FGDs.

2.3 GFDs were conducted with 19 participants.

a.How many people formed a group?

b.Were males and females brought together for the discussions?

3.In the analysis paragraph, there is a statement that reads “The codebook was discussed with the second author and after the agreement, the codebook was then in NVIVO version 11 software before the coding” is not clear. The authors need to revise this statement.

4.State the epistemological approach adopted for the work. Was it interpretative or phenomenological, inductive or deductive? The authors need to have a statement to clarify this.

5.Also clarify how socially desirable responses, confidentiality and privacy was ensured in this study.

a.How was the informed consent obtained for the FGDs? Was it a group or individual consent?

6.There is need for proof reading to correct grammatical errors.

7. PLOS authors have the option to publish the peer review history of their article (what does this mean?). If published, this will include your full peer review and any attached files.

Reviewer #3: **Yes: **Samuel Tamti Chatio

---

## [Author Response · Author response to Decision Letter 1]

14 Jan 2021

All responses to each comments are included in the matrix

---

## [Editor Report · Decision Letter 2]

18 Jan 2021

It is beyond remuneration: Bottom-up health workers’ retention strategies at the primary health care system in Tanzania

PONE-D-20-09192R2

Dear Dr. Sirili,

We’re pleased to inform you that your manuscript has been judged scientifically suitable for publication and will be formally accepted for publication once it meets all outstanding technical requirements.

Kind regards,

Khin Thet Wai, MBBS, MPH, MA (Population & Family Planning Res.)

Academic Editor

PLOS ONE

Additional Editor Comments (optional):

All comments are fully addressed.
---

## [Editor Report · Acceptance letter]

30 Mar 2021

PONE-D-20-09192R2 

It is beyond remuneration: Bottom-up health workers’ retention strategies at the primary health care system in Tanzania 

Dear Dr. Sirili:

I'm pleased to inform you that your manuscript has been deemed suitable for publication in PLOS ONE. Congratulations! Your manuscript is now with our production department. 

Kind regards, 

on behalf of

Dr. Khin Thet Wai 

Academic Editor

PLOS ONE